# Stability Characterization of the Novel Anti-Cancer HM-10/10 HDL-Mimetic Peptide

**DOI:** 10.3390/ijms241210054

**Published:** 2023-06-13

**Authors:** Michael P. Dempsey, Katelyn E. Andersen, Brittney M. Wells, Mitchell A. Taylor, Clay L. Cashman, Lesley B. Conrad, Claire A. Kearney, Mary B. Conklin, Emily R. Via, Emily M. Doe, Ravikiran Komirisetty, Susan Dearborn, Srinivasa T. Reddy, Robin Farias-Eisner

**Affiliations:** 1School of Medicine, Creighton University, Omaha, NE 68178, USA; dempseymp@gmail.com (M.P.D.); katelynandersen@gmail.com (K.E.A.); brittneywells@creighton.edu (B.M.W.); mitchelltaylor1@creighton.edu (M.A.T.); claycashman@creighton.edu (C.L.C.); lesleyconrad@creighton.edu (L.B.C.); clairekearney@creighton.edu (C.A.K.); mc2636@rwjms.rutgers.edu (M.B.C.); emily.via@stonybrookmedicine.edu (E.R.V.); emilydoe@creighton.edu (E.M.D.); 2David Geffen School of Medicine, The University of California at Los Angeles, Los Angeles, CA 90095, USA; 3Lynch Comprehensive Cancer Research Center, School of Medicine, Creighton University, Omaha, NE 68178, USA; 4Office of the President, Western University of Health Sciences, Pomona, CA 91766, USA; 5Department of Medicine, Division of Cardiology, David Geffen School of Medicine, The University of California at Los Angeles, Los Angeles, CA 90095, USA; rkomirisetty@mednet.ucla.edu (R.K.); sreddy@mednet.ucla.edu (S.T.R.); 6Charles River Laboratories International, Stone Ridge, NY 12484, USA; susan.dearborn@crl.com; 7Department of Molecular and Medical Pharmacology, David Geffen School of Medicine, The University of California at Los Angeles, Los Angeles, CA 90095, USA; 8Jonsson Comprehensive Cancer Center, The University of California at Los Angeles, Los Angeles, CA 90095, USA; 9Interdepartmental Program in Molecular Toxicology, School of Public Health, University of California at Los Angeles, Los Angeles, CA 90095, USA; 10California NanoSystems Institute, University of California at Los Angeles, Los Angeles, CA 90095, USA; 11Office of the President, Western University of Health Sciences, Lebanon, OR 97355, USA

**Keywords:** ovarian epithelial adenocarcinoma, colonic epithelial adenocarcinoma, novel cancer treatment, drug innovation, stability

## Abstract

Epithelial adenocarcinoma of the ovary and colon are associated with the highest rates of cancer-related deaths in women in the U.S. The literature supports the role of HDL-associated apolipoproteins in the treatment of cancer and other pro-inflammatory diseases. Previously, we developed a novel 20-amino acid mimetic peptide, HM-10/10, which potently inhibits tumor development and growth in colon and ovarian cancer. Here, we report the properties of HM-10/10 relative to its stability in vitro. The results demonstrated that HM-10/10 had the highest half-life in human plasma compared to plasma from other species tested. HM-10/10 demonstrated stability in human plasma and simulated gastric environment, increasing its promise as an oral pharmaceutical. However, under conditions modeling the small intestine, HM-10/10 demonstrated significant degradation, likely due to the peptidases encountered therein. Furthermore, HM-10/10 demonstrated no evidence of time-dependent drug–drug interactions, although it demonstrated CYP450 induction slightly above cutoff. As proteolytic degradation is a common limitation of peptide-based therapeutics, we are pursuing strategies to improve the stability properties of HM-10/10 by extending its bioavailability while retaining its low toxicity profile. HM-10/10 holds promise as a new agent to address the international women’s health crisis of epithelial carcinomas of the ovary and colon.

## 1. Introduction

Epithelial adenocarcinoma of the ovary (EOC) and colon (CC) are associated with the highest rates of cancer-related deaths in women in the U.S [1,2]. Unfortunately, most patients have advanced-stage disease at the time of initial presentation to a clinician [3]. Disease progression is inevitable in over 50% of such patients, with the emergence of chemotherapy-resistant and refractory disease in the majority. Over 70% of patients eventually succumb to their disease when diagnosed in the advanced stages [4,5].

High density lipoprotein (HDL) is an important mediator of lipid homeostasis. HDL and HDL-associated molecules provide protective functions, including anti-inflammatory, antioxidant, anti-cancer, anti-microbial, and innate immunity in multiple cell types and animal models [6,7,8,9,10]. HDL mimetic peptides have shown efficacy in animal cancer models and demonstrate properties that make them attractive as potential anti-cancer agents [6,7,8,9,10]. We developed a novel chimeric high-density lipoprotein mimetic peptide named HM-10/10. The arginine-rich cationic domain of human apoE {([141–150] hApoE) L-R-K-L-R-K-R-L-L-R} was added to an apoJ mimetic named G * peptide {L-V-G-R-Q-L-E-E-F-L, corresponding to amino acids 113 to 122 in apoJ ([113–122] apoJ)} to form the HM-10/10 peptide, containing 20 amino acids with the sequence L-R-K-L-R-K-R-L-L-R-L-V-G-R-Q-L-E-E-F-L.

In previous studies, we examined three dual-domain HDL mimetic peptides, namely, AEM-28 (containing an 18-amino acid [AA] peptide from apoA-I and a 10-AA peptide from apoE), AEM-28-2 (an analog of AEM-28 that was developed to increase potency and limit local irritation that included the replacement of Lys by Arg and the addition of Ac-Aha- (Aha = α-aminohenoic acid) to the N-terminus instead of Ac- to enhance lipid-binding) and HM-10/10 (containing a 10-AA peptide from apoE and a 10-AA peptide of apoJ) [11,12]. The results demonstrated that the AEM-28 and AEM-28-2 peptides significantly decrease tumor development following CT26 cell injection [11]. AEM-28 and AEM-28-2 inhibited the viability of the chemotherapy-resistant human cancer cell lines OV2008 and SKOV3 and reduced cell viability while inducing apoptosis in CT26 cells in vitro.

We demonstrated that the novel 20-amino acid mimetic peptide HM-10/10 potently inhibits tumor development and growth in both colon and ovarian cancer models in vitro and in vivo. Beyond its potent anti-cancer properties, we recently reported that HM-10/10 protects against chemically induced macular degeneration in mice [12], supporting the role of HM-10/10 as an anti-inflammatory agent. These findings collectively support our goal of developing HM-10/10 as our apolipoprotein mimetic peptide candidate of choice for further characterization and to advance its use as a novel therapeutic agent against pro-inflammatory conditions, including cancers such as EOC and CC.

Here, we report on the stability characterization of HM-10/10. The purpose of this study was to determine the absorption, distribution, metabolism, and excretion (ADME) attributes of HM-10/10. More specifically, we investigated HM-10/10’s stability and intrinsic clearance potential, protein binding, and CYP450 induction and inhibition, indicators of potential drug interactions. These studies were conducted in the following animal and human matrices: CD-1 mouse, Sprague-Dawley rat, beagle dog, cynomolgus monkey, human plasma (K_2_EDTA, obtained as previously frozen), and simulated gastric and intestinal fluids. In addition, HM-10/10 stability was determined in potassium phosphate buffer at three different pH levels. The ADME assays conducted on the selected species are critical in gaining insight into metabolism and potential drug interactions and contributed significantly to construction of the overall drug discovery and development strategy early in the process in order to de-risk potential poor outcomes while maximizing the overall safety and efficacy profile. The successful results from the stability characteristic studies support further development of HM-10/10 using a novel enhanced delivery vehicle to overcome the limitations encountered in the HM-10/10 mimetic peptide native state.

## 2. Results

### 2.1. HM-10/10 Demonstrates Stability in Plasma Matrices

HM-10/10 and the control articles lovastatin and propantheline were analyzed for stability in human, CD-1 mouse, Sprague-Dawley rat, Beagle dog, and Cynomolgus monkey plasma matrices over 120 min. HM-10/10 was shown to have a half-life above 50 min in all matrices tested except CD-1 mouse plasma (Figure 1, Table 1). The control articles demonstrated a wide range of half-lives (Table 1). Samples were tested at 2 μM.

### 2.2. HM-10/10 Demonstrates Stability in the Denaturing Environment of Gastric Fluid but Degrades Quickly under Proteolytic Intestinal Conditions

When developing any potentially oral-route pharmaceutical agent, the denaturing and pepsin-rich environment of the stomach must be considered. The stability of HM-10/10 was tested in simulated gastric fluid (SGF) and simulated intestinal fluid (SIF) over 120 min. HM-10/10 demonstrated excellent resilience under SGF conditions, but degraded quickly under SIF conditions (Table 2, Figure 2). After 120 min under SGF conditions, 71.8% of HM-10/10 was detected, while after only 20 min under SIF conditions, no HM-10/10 could be detected (Table 2, Figure 2). Samples were tested at 2 μM.

### 2.3. HM-10/10 Demonstrates Stability under Acidic and Physiological pH Conditions

The stability of HM-10/10 under varying pH conditions was determined by adding the compound to potassium phosphate buffers of pH 1.3 (acidic), 5.5 (slightly acidic), and 7.4 (physiological pH), with 7-Ethoxycoumarin used as a control article. Under acidic and slightly acidic conditions, the half-life of HM-10/10 could not be determined, as the percentage of HM-10/10 was higher at the end of the time course than at the beginning (Table 3, Figure 3). HM-10/10 demonstrated stability under acidic and physiological pH conditions, with all half-lives above 120 min and greater than 80% of the compound remaining after 60 min (Table 3, Figure 3). Samples were tested at 2 µM.

### 2.4. HM-10/10 Shows Low Clearance in Liver Microsomes

In the metabolizing enzyme-specific environment of human liver microsomes, HM-10/10 demonstrated good stability. In human liver microsomes, the half-life estimate was 20.3 min, while the intrinsic clearance was calculated to be 61.5 mL/min/kg (Table 4, Figure 4). Verapamil, the control article, demonstrated a half-life of 13.8 min with an intrinsic clearance of 90.5 mL/min/kg in human liver microsomes (Table 4, Figure 4). The full range of HM-10/10 half-lives in the animal models did not vary significantly between animal models, with the human liver microsomes demonstrating the lowest half-life at 20.3 min and the beagle dog liver microsomes demonstrating the greatest half-life at 33.0 min (Table 4, Figure 4). The cynomolgus monkey liver microsome results were most similar to those for the human liver microsomes, with a half-life of 21.8 min and an intrinsic clearance of 85.8 mL/min/kg (Table 4, Figure 4).

### 2.5. HM-10/10 Stability and Intrinsic Clearance in Hepatocytes

The half-life and intrinsic clearance values of HM-10/10 were investigated in hepatocytes of the same array of animal models used for plasma stability. Verapamil was used as a control article. The stability of HM-10/10 in human hepatocytes was the median value across the five different cell cultures, with 2.5% remaining after incubation for 30 min. This led to a half-life estimate of 5.28 min and an intrinsic clearance value of 708 mL/min/kg (Table 5). The Sprague-Dawley rat culture had the closest half-life estimate to the human cell culture, at 5.81 min with an intrinsic clearance value of 1260 mL/min/kg (Table 5). Verapamil returned higher half-life estimates and intrinsic clearance values relative to HM-10/10 in all hepatocyte cultures except for the Cynomolgus monkey, in which they were identical with those of HM-10/10 (Table 5).

### 2.6. Binding of HM-10/10 to Proteins

The binding of HM-10/10 to proteins in CD-1 mouse, Sprague-Dawley rat, beagle dog, cynomolgus monkey, and human plasma was successfully determined (Table 6). The percentage bound to plasma proteins after 2.5 h at 37 °C ranged from 15.3% (mouse) to 79.7% (human). The stability of HM-10/10 in each matrix under the assay conditions ranged from 14.6% (mouse) to 88.5% (human). The control article (Warfarin) performed as expected in all species tested, confirming assay performance.

### 2.7. HM-10/10 Time-Dependent Inhibition (TDI) of CYP450 Isoenzymes in Hepatocytes

HM-10/10 and the control compounds fluvoxamine, ticlopidine, quercetin, sulfaphenazole, omeprazole, paroxetine, and mifepristone were analyzed at concentrations up to 50 µM for inhibition of cytochrome P450 (CYP450) isoenzymes. Inhibition was tested for HM-10/10 and various CYP450 isozymes under the following conditions: +/− NADPH, both with and without a 30-min preincubation with microsomes (enzyme source) (PI). Inhibition of CYP450 isoenzymes by 50µM HM-10/10 under these different conditions ranged from −26.4 to 42.2% compared to control time-dependent inhibitors, which exhibited >85.4% inhibition at maximum test concentrations (Table 7). These values indicate that HM-10/10 showed no TDI under the tested conditions. A negative TDI indicates that the test article in question will not result in DDI when taken with other medications by potentially altering the CYP450 pathway being used to metabolize a co-administered drug. The control inhibitors yielded half-maximal inhibitory concentrations (IC_50_), which is consistent with historical data.

### 2.8. HM-10/10 Drug–Drug Induction (DDI) of Three Different CYP450 Isoenzymes

In this assay, induction of three different CYP450 isoenzymes (1A2, 2B6, and 3A4) in single-donor lots of human hepatocytes (HH1086, HH1142, and HH1144) by HM-10/10 at 2, 10, and 25 µM was observed in at least one of the three single-donor lots of hepatocytes based on increases in enzyme activity (Table 8) or mRNA expression that were ≥2-fold the vehicle control response (Table 9).

The induction of CYP1A2 as measured by enzyme activity was observed in three single-donor hepatocyte lots tested at 25 µM HM-10/10; however, it was not dose-dependent, with induction at only two of three single-donor hepatocyte lots tested at 10 µM of HM-10/10 and no induction of single-donor hepatocyte lots at 2 µM of HM-10/10 (Table 6). At a concentration of 2 µM HM-10/10, based on fold increases in mRNA expression there was minimal induction of CYP1A2 (2.24-fold increase) observed in one of three single-donor hepatocyte lots.

The most significant induction was observed with CYP2B6, which was induced by HM-10/10 at 2, 10, and 25 µM in the three single-donor lots of hepatocytes. This induction appears concentration-dependent, with mean fold-increases in enzyme activity of 3.00, 4.96, and 6.27 at 2 µM, 10 µM, and 25 µM HM-10/10, respectively (Table 8). As measured by fold increase in mRNA expression, induction did not appear to be concentration-dependent, with observed increases in three of three single-donor hepatocyte lots at concentrations of 2 µM and 10 µM, and in only one of three single-donor hepatocyte lots at 25 µM of HM-10/10 (Table 9).

The induction of CYP3A4 as measured by enzyme activity was observed with HM-10/10 at 2, 10, and 25 µM HM-10/10 in at least one of the three single-donor lots of hepatocytes. The observed fold increases in enzyme activity were minimal, and were not concentration-dependent (Table 8). As measured by fold increase in mRNA expression, induction was greatest overall for CYP3A4 in comparison to CYP1A2 and CYP2B6. However, induction was observed in the same two of three single-donor hepatocyte lots at all concentrations of HM-10/10 (2 µM, 10 µM, and 25 µM), respectively, while consistently not being seen in the HH1144 hepatocyte lot across all concentrations (Table 9). These positive in vitro results for CYP3A4 indicate the need to perform follow-up induction studies for CYP2C8, CYP2C9, and CYP2C19, as these isoforms are also induced via activation of the pregnane X receptor (PXR).

Across the three single-donor lots of hepatocytes, the average stability of HM-10/10 over 24 h incubation with hepatocytes for both days of incubation was 0.6% at 2 µM, 0.7% at 10 µM, and 0.3% at 25 µM (Table 8).

## 3. Materials and Methods

### 3.1. HM-10/10 and Plasma Stock Solution Preparation

A test stock solution of HM-10/10 at 1 mM in 50:50 (*v*:*v*) ethanol:water was prepared and sent to Charles River Laboratories, Inc. for stability analysis. Synthesis of HM-10/10 adhered to FDA GLP compliance protocols for purity and quality assurance. Briefly, HDL mimetic HM-10/10 peptide (L-R-K-L-R-K-R-L-L-R-L-V-G-R-Q-L-E-E-F-L) was synthesized from all L-amino acids (LifeTein, LLC, Hillsborough, NJ, USA). The peptide had >98% purity as determined by high-performance liquid chromatography (HPLC) and mass spectrometry (MS). The peptide was dissolved in H_2_O for cell culture.

Control Stock Preparation: Individual control article stock solutions of 7-Ethoxycoumarin, lovastatin, and propantheline were prepared at a concentration of 10 mM in DMSO and were further diluted to 1 mM with DMSO.

### 3.2. Liver Microsome Stock Solution Preparation

Frozen pooled liver microsomes (CD-1 mouse, Sprague-Dawley rat, beagle dog, cynomolgus monkey, and human) were purchased from Sekisui XenoTech, LLC (Kansas City, KS, USA). (The following information see Appendix A).

HM-10/10 Stock Preparation: A stock solution of HM-10/10 was received at 1 mM in 50:50 (*v*:*v*) ethanol:water. Synthesis of HM-10/10 adhered to FDA GLP compliance protocols for purity and quality assurance.

Control Stock Preparation: A stock solution of verapamil was prepared at 10 mM in DMSO and was diluted to 2 mM and 1 mM in DMSO as applicable.

Control and test articles (1 mM stock solutions) were diluted ten-fold with acetonitrile to 0.1 mM, followed by further 1:50 dilution into an NADPH solution at 2.0 mM (2X final assay concentration) in warm (37 °C) 0.1 M potassium phosphate buffer, pH 7.4.

### 3.3. Hepatocyte Stock Solution Preparation

Frozen Hepatocytes (CD-1 mouse, Sprague-Dawley rat, beagle dog, cynomolgus monkey, and human) were purchased from BioIVT (Westbury, NY, USA). (The following information see Appendix A).

HM-10/10 Stock Preparation: A stock solution of HM-10/10 was received at 1 mM in 50:50 (*v*:*v*) ethanol:water. Synthesis of HM-10/10 adhered to FDA GLP compliance protocols for purity and quality assurance.

Control Stock Preparation: A stock solution of verapamil was prepared at 2 mM in DMSO and was further diluted to 1 mM with DMSO.

Just prior to the assay, the stocks were diluted 1:10 with acetonitrile, followed by further 1:50 dilution into InVitroGRO KHB buffer (Celsis, Catalog No. Z99074).

### 3.4. Protein Binding Stock Solution Preparation

HM-10/10 Stock Preparation: A stock solution of HM-10/10 at 1 mM in 50:50 (*v*:*v*) ethanol:water was prepared and sent to Charles River Laboratories, Inc. (Worcester, Mass, USA) for protein binding analysis. Synthesis of HM-10/10 adhered to FDA GLP compliance protocols for purity and quality assurance.

Control Stock Preparation: Stock solutions of 7-Ethoxycoumarin and Warfarin were prepared at a concentration of 10 mM in DMSO and further diluted to 1 mM with DMSO.

### 3.5. Plasma Drug Stability Assay Procedure

Frozen plasma matrices were thawed and centrifuged at 3100 rpm (2074× *g*) for 5 to 10 min at 4 °C. The supernatant was removed and incubated at 37 °C for at least 10 min before use. The pH of each matrix was checked and adjusted to pH 7.4 using 10% phosphoric acid or 1 N sodium hydroxide as necessary.

Test and control article stock solutions were spiked (1.5 µL) into matrix (0.749 mL) to a final assay concentration of 2 µM and mixed. Immediately after mixing, duplicate aliquots of compound-spiked plasma were transferred to matrix tubes in 96-well plates for timepoints of 0, 10, 20, 30, 60, and 120 min. These tubes were sealed and incubated at 37 °C with shaking.

At each timepoint, the corresponding tubes containing samples were quenched by the addition of six volumes of ice-cold 0.1% perchloric acid in methanol containing the internal standards. The quenched samples were mixed briefly by vortexing and stored in a −20 °C freezer.

After the final timepoint (120 min), the plates with quenched samples were centrifuged at 3100 rpm (2000× *g*) for 10 min at 4 °C to sediment the precipitated protein. Next, an aliquot of the supernatant was transferred to a new 96-well microtiter plate and diluted with an equal volume of 100:0.1 (*v*:*v*) water:formic acid. Sample plates were stored refrigerated until analysis.

### 3.6. Liver Microsome Drug Stability Assay Procedure

Frozen pooled liver microsomes (purchased from Sekisui XenoTech, LLC), (Kansas City, KS, USA). were quickly thawed in a 37 °C water bath and placed on ice. The liver microsomes (20 mg protein/mL) were diluted in 0.1 M potassium phosphate buffer, pH 7.4, (warmed to 37 °C) to a concentration of 1 mg protein/mL (2X final assay concentration). (The following information see Appendix A).

To initiate the reaction, 150 µL of diluted 2X microsomes were added to an equal volume of 2X compound/NADPH solution in a polypropylene 96-well microtiter plate. The plate was incubated with gentle shaking at 37 °C. Duplicate 30 µL aliquots were removed immediately after compound addition (time zero) and at assay timepoints of 15, 30, 60, 90, and 120 min.

At each timepoint, control and test article samples were quenched with 180 µL of ice-cold 0.1% perchloric acid in methanol containing internal standards. The quenched samples were vortex-mixed for at least 1 min and were stored in a −20 °C freezer.

After the final timepoint (120 min), the quenched samples were vortexed (1 min) and then centrifuged at 3100 rpm for 10 min at 4 °C. An aliquot of the supernatant was removed, transferred to a new 96-well plate, and diluted with an equal volume of 100:0.1 (*v*:*v*) water:formic acid. The sample plate was sealed, mixed, and then refrigerated for storage until analysis.

### 3.7. Hepatocyte Drug Stability Assay Procedure

Frozen hepatocytes (CD-1 mouse, Sprague-Dawley rat, beagle dog, cynomolgus monkey, and human) were purchased from BioIVT (Westbury, NY, USA). (The following information see Appendix A).

These were immediately thawed in a 37 °C water bath and rinsed with warm (37 °C) InVitroGRO HT Medium (Catalog No. Z99019). The hepatocytes were centrifuged at 100× *g* at 4 °C for 10 min. The cell pellets were resuspended in InVitroGRO KHB buffer. The cell count was adjusted to 1 million cells/mL (2X the desired assay concentration) with additional InVitroGRO KHB buffer.

To start the assay, an aliquot (125 µL) of 2X compound solution and an equal volume of 2X hepatocytes were combined in polypropylene 96-well plates (two wells per compound per matrix). The assay plates were incubated at 37 °C with 5% carbon dioxide. The final assay concentrations were 2 µM for HM-10/10, 2 µM for verapamil (control article) and 0.5 million cells/mL for hepatocytes.

Immediately after combining (time zero) and at assay timepoints (15, 30, 60, 120, and 180 min), an aliquot (30 µL) from each reaction well was transferred to a 96-well plate and quenched with 180 µL of ice-cold 0.1% perchloric acid in methanol containing the internal standards. The quenched samples were stored on ice/refrigerated. After the samples from the final timepoint were quenched, the samples were centrifuged at 3100 rpm for 10 min at 4 °C. An aliquot of the supernatant was transferred to a new plate and diluted with an equal volume of 100:0.1 (*v*:*v*) water:formic acid. The sample plate was sealed, mixed, and then refrigerated for storage until analysis.

### 3.8. Protein Binding Assay Procedure

Frozen plasma matrices were thawed and centrifuged at 3100 rpm (2000× *g*) for 5 min at 4 °C. The supernatant was removed and incubated at 37 °C for at least 10 min prior to use. The pH of each matrix was checked and adjusted to pH 7.4 using 10% phosphoric acid or 1N sodium hydroxide, if necessary. Test and control solutions were spiked (1.5 µL) into matrix (0.749 mL) to a final assay concentration of 2 µM. The final DMSO concentration in the matrix was 0.2%.

Compound-spiked matrices were incubated at 37 °C for 10 min. After incubation, aliquots of the compound-spiked matrix were removed and matrix-matched with an equal volume of phosphate buffered saline (PBS), then quenched by adding six volumes of ice-cold 0.1% perchloric acid in methanol. Quenched sample plates were sealed and stored refrigerated until the end of the assay.

After incubation to reach binding equilibrium, each spiked matrix was transferred into three polycarbonate ultracentrifuge tubes (0.5 mL per tube) and then placed into a Beckman TLA-100.4 rotor pre-warmed to 37 °C. Samples were centrifuged at 100,000 rpm (approximately 500,000× *g*) for 2.5 h at 37 °C with the lowest brake setting. Concurrently, the remaining compound-spiked matrices were incubated at 37 °C.

Following ultracentrifugation, any lipid layer was carefully removed and an aliquot of supernatant was transferred to a microtiter plate containing an equal volume of blank matrix for matrix-matching purposes. From concurrently incubated (non-centrifuged) spiked matrices, an aliquot was removed and matrix-matched with an equal volume of PBS. Matrix-matched samples were quenched by the addition of six volumes of ice-cold 0.1% perchloric acid in methanol, which contained internal standards, to precipitate the proteins. Quenched sample plates were sealed and refrigerated for storage.

Quenched samples were vortex-mixed and then centrifuged at 3100 rpm (2000× *g*) for 10 min at 4 °C to sediment the precipitated protein. An aliquot of the supernatant was transferred to a new 96-well microtiter plate and diluted with an equal volume of 100:0.1 (*v*:*v*) water:formic acid. Sample plates were refrigerated for storage until analysis.

### 3.9. Cytochrome P450 Time-Dependent Inhibition (TDI) Assay Procedure

All procedures were conducted in accordance with applicable Standard Operating Procedures (SOPs), the Testing Facility’s Discovery Guidelines, and good scientific practices. The data were reviewed for accuracy and consistency.

A test article (HM-10/10) stock solution was prepared at 5 mM in DMSO and further diluted to the following concentrations: 3.33, 2.22, 0.555, 0.139, 0.0348, 0.00870, 0.00218, 0.000545, and 0.0 mM using DMSO. The HM-10/10 solutions prepared by serial dilution in DMSO, as well as DMSO alone, were diluted 33.3-fold in warm (37 °C) 100 mM potassium phosphate buffer, pH 7.4, both with and without 3 mM NADPH to make 3X compound solutions in buffer with and without NADPH.

Stock solutions of the positive control inhibitors (fluvoxamine, ticlopidine, quercetin, sulfaphenazole, omeprazole, paroxetine, and mifepristone) were prepared at 50 mM in DMSO and further diluted to the following concentrations: 33.3, 22.2, 5.55, 1.39, 0.348, 0.0870, 0.0218, 0.00545, and 0.0 mM using DMSO. Positive control inhibitor solutions prepared by serial dilution in DMSO, as well as DMSO alone, were diluted 33.3-fold in acetonitrile followed by a 100-fold dilution in warm (37 °C) 100 mM potassium phosphate buffer, pH 7.4, with and without 3 mM NADPH to make 3X compound solutions in buffer with and without NADPH.

The substrates in DMSO (40 mM phenacetin, 25 mM bupropion, 1 mM amodiaquine, 10 mM diclofenac, 40 mM mephenytoin, 10 mM dextromethorphan, 125 mM testosterone) and in methanol (3.06 mM midazolam) were added to warm (37 °C) 100 mM potassium phosphate buffer, pH 7.4, with 3 mM NADPH to make a (3X) buffer/cofactor/substrate (BCS) solution with concentrations ranging from 3 µM to 150 µM. Similar dilution of substrates was performed in warm (37 °C) 100 mM potassium phosphate buffer, pH 7.4, containing no NADPH to make a 3X buffer/substrate (BS) solution. These two dilution steps resulted in 3X substrate solutions in buffer with and without NADPH.

Pooled human liver microsomes were submerged in a 37 °C water bath until just thawed and then placed on ice. The microsomes were diluted in warm (37 °C) 100 mM potassium phosphate buffer, pH 7.4 to 0.3 mg/mL (3X solution), and warmed to 37 °C.

To initiate the reactions (in duplicate), 50-µL aliquots of 3X test/control articles, 3X human liver microsomes, and a 3X substrate (BCS or BS) solution were added together. For the pre-incubation assay condition, the assay plates were incubated at 37 °C for 30 min with a test article or control inhibitor and microsomes prior to the addition of the substrate solution. For HM-10/10 and the positive control inhibitors, final concentrations were 0.00 (DMSO), 0.00545, 0.0218, 0.0870, 0.348, 1.39, 5.55, 22.2, 33.3, and 50.0 µM.

For the T^0^ control (no enzyme activity) condition, 50-µL aliquots (from the samples containing 0 µM and the highest concentration of the test or control article) were immediately removed and quenched with 200 µL of ice-cold acetonitrile containing the internal standards (250 ng/mL of CYP-specific internal standards) in order to precipitate the proteins. The assay plates were sealed and incubated at 37 °C with very gentle agitation. After a 30-min incubation period, 50-µL aliquots were removed and quenched with 200 µL of ice-cold acetonitrile containing internal standards. The quenched samples were vortex-mixed and refrigerated for storage.

Quenched samples were centrifuged at 3100 rpm (2000× *g*) for 10 min at 4 °C. For each test or control article, supernatants from CYPs 1A2, 2B6, and 2C8 assays were pooled (50 μL per sample) in a new plate and supernatants from CYPs 2D6, 3A4 (midazolam substrate), and 3A4 (testosterone substrate) assays were pooled (50 μL per sample) in a new plate. The pooled samples were diluted with 300 μL water. For CYPs 2C9 and 2C19 assay samples, supernatant (50 μL) was transferred to a new plate and each aliquot was diluted with 100 μL water. Plates were sealed and refrigerated for storage until analysis.

The inhibition was measured by quantitating the metabolite generated by the CYP450 specific substrate being metabolized, which should decrease as the concentration of the test inhibitor increases if the test article is in fact inhibiting the particular CYP450 being tested.

### 3.10. Cytochrome P450 Drug-Drug Induction (DDI) Assay Procedure

#### 3.10.1. Enzyme Activity

On Day 1, three lots of pre-plated human hepatocytes (Catalog No. 82043-96OD), one plate per lot, were obtained from In Vitro ADMET Laboratories (Malden, MA, USA). The transport medium was replaced with 200 μL of warm (37 °C) incubation medium (InVitroGRO HI, BioIVT Catalog No. Z99009) and the plate was placed in a humidified 5% carbon dioxide incubator at 37 °C for 24 h.

Positive control inducers were dissolved in dimethyl sulfoxide (DMSO) at 50 mM omeprazole, 1000 mM phenobarbital, and 20 mM rifampicin. Substrates were dissolved in DMSO at 100 mM phenacetin, 150 mM bupropion, and 125 mM testosterone.

On Day 2, HM-10/10 test solutions were freshly prepared by combining 500 µM HM-10/10 in phosphate-buffered saline (PBS), PBS, and warm (37 °C) InVitroGRO HI Medium for final assay concentrations of 25, 10, and 2 µM HM-10/10 in 5:95 (*v*:*v*) PBS:InVitroGRO HI Medium. In addition, test article vehicle (PBS) was added to warm (37°C) InVitroGRO HI Medium in a 5:95 (*v*:*v*) ratio. Inducer control solutions were freshly prepared by diluting the stock solutions 1000-fold with warm (37 °C) InVitroGRO HI Medium to 50 µM omeprazole, 1000 µM phenobarbital, and 20 µM rifampicin (0.15% DMSO). The vehicle control was prepared by diluting 3 µL DMSO in warm (37 °C) InVitroGRO HI Medium (1997 µL). The plates were removed from the incubator and the hepatocytes were prepared for the induction assay by replacing the cell culture medium with medium containing either the test article, a control inducer, or vehicle control (200 µL per well). The hepatocyte plates were returned to the cell culture incubator and incubated for 24 h.

For each 24 h incubation, 50 µL of each medium containing the test article (HM-10/10) were collected at the start (T_0_) and end of 24 h incubation (T_24_) and quenched with 300 µL of 1% perchloric acid in methanol and 50 µL of internal standard cocktail (carbutamide, chrysin, and glyburide each at 1.71 µg/mL water). These samples were analyzed to test the stability of HM-10/10 after 24 h incubation with the hepatocytes. The quenched sample plates were stored covered in a −20 °C freezer.

On Day 3, induction was repeated following the same steps as described above for the previous day, and plates were returned to the cell culture incubator for an additional 24 h.

On Day 4, CYP substrate stocks were diluted 1000-fold in warm (37 °C) assay buffer (InVitroGRO KHB; BioIVT Catalog No. Z99074) for a nominal final concentration of 100 µM phenacetin, 150 µM bupropion, and 125 µM testosterone. After aliquots were taken to measure test article stability, all the remaining medium from each well in the hepatocyte plate was removed, all wells were washed once using 200 μL of blank assay buffer, and then 100 μL of the 1000-fold diluted substrate solutions were added to the appropriate wells. The assay plate was returned to the humidified 5% carbon dioxide incubator at 37 °C for 1 h incubation.

During incubation, lysis buffer was freshly prepared by adding 600 µL of 2-mercaptoethanol to 59.4 mL of Buffer RLT (QIAGEN catalog No. 1015762). At the end of the incubation time, 100 μL of mRNA lysis buffer was added to each well to stop the reaction. Fifty microliters of this reaction mixture were transferred into a new plate and quenched with 200 μL of ice-cold acetonitrile containing a CYP-specific stable labeled internal standard (250 ng/mL acetaminophen-d4, 250 ng/mL hydroxybupropion-d6, or 250 ng/mL hydroxytestosterone-d3). After removal of the aliquots from the plate for enzyme activity determination, the assay plates were sealed and stored in a −70 °C freezer until use for RT-PCR, mRNA analysis.

The quenched samples were centrifuged at 3100 rpm (2000× *g*) for 5 min at 4 °C. For the induction assay samples, an aliquot (50 µL) of the supernatant was transferred to a new plate and diluted with 100 µL Milli-Q water. For the test article stability samples, an aliquot (50 µL) of the supernatant was transferred to a new plate and diluted with 60 µL 0.1% formic acid in Milli-Q water. These samples for the enzyme activity assay and article stability test were refrigerated for storage until analysis.

#### 3.10.2. Messenger RNA (mRNA) Expression and Analysis

RNA isolation and quantification: E-Z 96 Total RNA Kit (Omega Bio-tek, Catalog No. R1034-02) was used for RNA isolation and Quant-iT RNA assay kit (ThermoFisher, Catalog No. Q33140) was used for RNA quantification. The plates were removed from a −80 °C freezer and equilibrated to room temperature. RNA was extracted using the manufacturer’s recommended protocol. Eighty-one microliters of 100% ethanol were added into each well containing 150 µL of lysed sample to make a final ethanol concentration at 35%.

cDNA generation: For cDNA generation, a High-Capacity cDNA Reverse Transcription Kit (Applied Biosystems, Catalog No. 4319983) and QuantStudio 3 PCR system (Applied Biosystems) were used. In this step, 2X Reverse Transcription Master Mix was prepared using the contents of the kit. Then, 10 µL of total RNA was added to 10 µL of 2X Reverse Transcription Master Mix in a microplate and the plate was sealed. This plate was then run in a thermocycler to generate cDNA.

Real time PCR: SsoAdvanced Universal SYBR^®^ Green Supermix (BIO-RAD, Catalog No. 172-5270) and the QuantStudio 3 Real-Time PCR system (Applied Biosystems) were used to perform real-time PCR on the generated cDNA. In this step, a PCR mix was prepared using PCR Supermix, SYBR^®^ primers for the targeted genes, and nuclease-free water. Two microliters of ten-fold diluted cDNA samples (5 µL of cDNA into 45 μL of nuclease-free water) were added to 8 µL of PCR Mix in a microplate and sealed. This plate was then run through the Real-Time PCR system to generate the mRNA expression data.

#### 3.10.3. SYBR Primers (BIO-RAD)

Human GAPDHF: AAGGTGAAGGTCGGAGTCAAR: AATGAAGGGGTCATTGATGGHuman CYP1A2Unique Assay ID: qHsaCID0015160Chromosome Location: 15:75045562-75047256Human CYP2B6F: CCCTTTTGGGAAACCTTCTGR: GTCCCAGGTGTACCGTGAAGHuman CYP3A4F: TTTTGTCCTACCATAAGGGCTTTR: CACAGGCTGTTGACCATCAT

### 3.11. Bioanalysis

For HM-10/10 and propantheline, a Waters XSELECT HSS T3 2.5 µm 30 × 2.1 mm column was used with a gradient (0.9 mL/minute flow rate) starting at 99% mobile phase A (0.1% formic acid in water) to 95% mobile phase B (0.1% formic acid in acetonitrile). For lovastatin, a Waters XSELECT HSS T3 2.5 µm 30 × 2.1 mm column was used with a gradient (1.2 mL/minute flow rate) starting at 70% mobile phase A (0.1% formic acid in water) to 95% mobile phase B (0.1% formic acid in acetonitrile). The column was set to a temperature of 55 °C. Analytes and internal standards were detected using an Applied Biosystems Sciex API-5500 triple quadrupole mass spectrometer with Agilent 1260 Infinity Binary Pump and Apricot Designs ADDA High-Speed Dual Arm Autosampling System. The instrument was equipped with an electrospray ionization source (500 °C) operated in positive-ion mode.

For HM-10/10 and 7-Ethoxycoumarin, a Waters XSELECT HSS T3 2.5 µm 30 × 2.1 mm column was used with a gradient (0.9 mL/minute flow rate) starting at 99% mobile phase A (0.1% formic acid in water) to 95% mobile phase B (0.1% formic acid in acetonitrile). The column was set to a temperature of 55 °C. Analytes and internal standards were detected using an Applied Biosystems Sciex API-5500 triple quadrupole mass spectrometer with Waters Acquity UPLC System. The instrument was equipped with an electrospray ionization source (600 °C) operated in the positive-ion mode.

Analytes and internal standards were monitored in multiple-reaction-monitoring scan mode using the previously optimized parent ion/product ion transitions “Q1/Q3” (Table 10).

### 3.12. Data Analysis and Calculations

Data were captured and processed using Sciex Analyst. Data were analyzed and results calculated using Microsoft Excel.

Peak area ratios for IS-normalized test article and control article counts were used to calculate the percent remaining relative to the time zero (T_0_) value. As appropriate, results were plotted and further analyzed to calculate half-life (T_1/2_) values.

Mean peak ratios were used to calculate the percent remaining relative to the T_0_ value: % Remaining at Time_x_ = (mean peak area ratio T_x_/mean peak ratio T_0_) × 100.

Results were plotted (ln (% remaining) vs. incubation time) and further analyzed to calculate half-life (T_1/2_) values: T_1/2_ = −0.693/Slope.

For the HM-10/10 stability assay in simulated gastric and intestinal fluids, propantheline was used to confirm retention time and carbutamide was used as the internal standard for calculations.

For the liver microsome trials, the HM-10/10 and verapamil intrinsic clearance values were calculated by the following equation: CL_int_ = (ln^2^/T_1/2_) × (1/protein conc.) × 1000.

For the cryopreserved hepatocyte trials, the HM-10/10 and verapamil intrinsic clearance values were calculated by the following equation: CL_int_ = (ln^2^/T_1/2_) × (1/million cells/mL) × (million cells/g liver) × (g liver/kg body weight).

For protein binding trials, the following calculations were made:$$\%\;Free=\frac{Peak\;Area\;Ratio\;of\;Supernatant}{Peak\;Area\;Ratio\;of\;Total}\times 100$$
$$\text{The}\pm \text{value}\;\text{for}\;\text{calculated}\;\%\;\text{Free}=\;Mean\;\%\;Free\times \sqrt{{\left(\frac{SD\;of\;Mean\;Peak\;Area\;Ratio\;of\;Supernatant}{Mean\;Peak\;Area\;Ratio\;of\;Supernatant}\right)}^{2}+{\left(\frac{SD\;of\;Mean\;Peak\;Area\;Ratio\;of\;Total}{Mean\;Peak\;Area\;Ratio\;of\;Total}\right)}^{2}}$$
$$Fu(Unbound\;fraction)=\frac{Peak\;Area\;Ratio\;of\;Supernatant}{Peak\;Area\;Ratio\;of\;Total}$$
% Bound=100%−% Free
$$\%\;Stability=\frac{Mean\;Peak\;Area\;Ratio\;after\;Incubation}{Mean\;Peak\;Area\;Ratio\;T0}\times 100$$

## 4. Discussion

The use of therapeutic peptides for cancer treatment has been broadly evaluated and has demonstrated promising outcomes; however, the effectiveness of use has selective requirements [13,14,15,16,17,18,19]. It has been reported by Smith et al. that the desired half-life range of an oral pharmaceutical is within 12–48 h [20]. This allows a length of time sufficient to achieve preferable once-daily dosing, but not so long that the potential of toxicity becomes an issue. HM-10/10 demonstrated a half-life of 164 min, or 2 h and 44 min, in human plasma. The half-life of HM-10/10 in human plasma is well below the range where toxicity poses a potential issue. It does fall short of the estimated range for a drug to achieve a once-daily dosing schedule; however, the efficacy of the peptide could very well overshadow the minor drawback of multiple daily doses. Additionally, the relatively shorter half-life could allow for more control over the therapeutic window. The half-life of HM-10/10 in Sprague-Dawley rat plasma was the closest to human plasma at 82.5 min, or 1 h and 22.5 min. This result supports the Sprague-Dawley rat as a good candidate for further in vivo studies of this peptide.

HM-10/10 showed remarkable stability under the acidic conditions of the stomach. In simulated gastric fluid, the peptide demonstrated a half-life of 372 min, or 6 h and 12 min. This is likely due to the higher ratio of positively charged amino acids to hydrophobic amino acids and the propensity of pepsin to cleave peptides higher in hydrophobic amino acids [21]. With the harsh environment of the stomach being a major hurdle in oral peptide absorption, this resilience is of great assistance to future use of HM-10/10 orally. However, the stability of the peptide in simulated intestinal fluid was significantly lower. The half-life of HM-10/10 in simulated intestinal fluid was 2.72 min, and 0% of the peptide remained after 60 min. This is not necessarily surprising considering the more diverse array of peptidases encountered in the small intestine but may pose a problem to the oral efficacy of the drug. Strategies such as PEGylation or substitution of certain residues with D-amino acids could be assessed in later studies to evaluate whether they can bypass the issue of intestinal degradation without compromising the efficacy of HM-10/10 [22]. Additionally, strategies that do not modify the peptide could be employed, such as liposomal or nanosphere packaging.

The stability of HM-10/10 was tested under three different pH conditions, mimicking the range of acidities that may be encountered as the drug travels from the stomach to the bloodstream. HM-10/10 showed exceptional stability under acidic conditions. The half-life of the peptide could not be determined in solutions of pH 1.3 and 5.5 due to there being over 100% of the peptide remaining after 120 min. The excess of the peptide relative to the 2µM concentration added is likely due to an error in measuring the amount added; nonetheless, the remarkable observation of the experiment was the lack of degradation in both acidic environments. This finding implicates proteolytic enzymes, independent of pH change, as being the cause of the poor stability of HM-10/10 under SIF conditions. The peptide demonstrated its lowest half-life in the pH test series for the physiologic pH range (pH of 7.4), though this is only relative to its remarkable stability under acidic conditions. At a pH of 7., HM-10/10 had a half-life of 433 min, or 7 h and 13 min. As the half-life of the peptide in human plasma was 2 h and 44 min, it seems that plasma peptidases will be of greatest concern when assessing its half-life in vivo, rather than pH.

Observing the stability of a therapeutic peptide as it passes through the liver is crucial in directing how the drug will be delivered [23]. Enzymes such as cytochrome P450s and UDP-glucuronosyltransferases contribute to what is called the first pass effect and can “inactivate” a significant portion of an oral drug before it has a chance to act upon its target. HM-10/10 returned results that suggest it may be significantly affected by the first pass effect, with a half-life estimate of 5.28 min and intrinsic clearance of 708 mL/min/kg in human hepatocytes. Sprague-Dawley rat hepatocytes had the closest half-life estimate and intrinsic clearance of HM-10/10 to human hepatocytes, again showing that this may be the best animal model for future in vivo studies. Interestingly, the human liver microsomes, an isolated version of the drug-metabolizing function of hepatocytes, returned an HM-10/10 half-life that was almost four times greater than the cryopreserved human hepatocytes. This could suggest that intracellular enzymes of human hepatocytes other than the known players in drug metabolism could be at work in the degradation of HM-10/10. These results add further evidence that techniques such as liposomal or nanoparticle packaging could be of great benefit to this peptide’s therapeutic potential.

Another important consideration in development of targeted drug therapies is the potential for drug–drug interactions (DDI), especially in cancer patients, who are commonly treated using multi-drug regimens. One of the most common mechanisms resulting in clinically relevant DDI is time-dependent inhibition (TDI) of CYP450 enzymes. This inhibition may be reversible or irreversible, with the latter being of particular concern, as de novo synthesis of the enzyme is required to restore activity. Results from HM-10/10 showed no TDI, which is a useful indicator of safety in drug development [24].

Induction of cytochrome P450 enzymes is associated with an increase of clinical drug–drug interactions (DDI). The clinical consequences of induction may include therapeutic failure caused by decreased systemic exposure of the drug itself or a co-administered therapy, or toxicity due to increased bioactivation. If a CYP enzyme is induced by a compound, this may increase the metabolism of a concurrent therapy of or itself (autoinduction), leading to a reduction in plasma levels and a potential decrease in drug efficacy. In addition, induction of CYP enzymes can result in toxicity by increasing reactive metabolite formation. Induction of CYP1A2, CYP2B6, and CYP3A4 gene expression can serve as sensitive representative endpoints for activation of AhR, CAR, and PXR respectively (i.e., nuclear receptor-mediated induction).

In assessing in vitro induction analysis, a cut-off is applied to fold induction observed based on FDA recommendations, i.e., concentration-dependent increase of mRNA expression with fold change ≥ 2-fold relative to vehicle control or increase >20% of positive control at expected hepatic concentrations of the drug. This provides a conservative potential risk of clinical DDI due to induction. If a positive result of ≥2-fold or increase >20% over positive control is observed in at least one donor hepatocyte, the compound should be considered an in vitro inducer and in vivo follow up studies may be required [25]. HM-10/10 demonstrated slight and therapeutically insignificant induction (based on a goal of an optimal therapeutic dose of ≤10 μΜ) on the CYP450 Induction assay in vitro. Consequently, further CYP450 Induction should be elucidated in vitro using the new enhanced HM-10/10 peptide (under development) and verified in vivo [26] if therapeutically significant induction is demonstrated.

Examining the binding of HM-10/10 to plasma proteins is another important measure for determining drug distribution and efficacy. Generally, only unbound therapeutic peptides are pharmacologically active and able to passively diffuse to target sites in the body [27,28]. To achieve desirable distribution of a drug, it is recommended that therapeutic peptides be less than 80–95% bound to plasma proteins [27]. Preliminary results from HM-10/10 protein binding studies demonstrate variable degrees of protein binding of HM-10/10 across organism plasma samples. While CD-1 mouse plasma yielded the lowest percentage of protein-bound HM-10/10 (15.3%), human plasma displayed the highest percentage of protein-bound HM-10/10 (79.7%). The high percentage of protein bound to HM-10/10 in human plasma suggests that HM-10/10 in its current state may not achieve optimal drug distribution throughout the body. These results further indicate that liposomal or nanoparticle packaging techniques may help to decrease protein binding and further improve the overall efficacy of HM-10/10.

Stability in both human plasma and the gastrointestinal tract is a well-established hurdle in the employment of novel peptide pharmaceuticals. An oral peptide must evade numerous proteolytic enzymes in the GI tract in order to be absorbed intact, and is then under constant attack by peptidases in the bloodstream. While novel strategies such as packaging peptides in liposomes or nanoparticle envelopes [29] are emerging as ways to circumvent this system, the pharmacokinetics of any potential therapeutic peptide must be assessed in numerous media before expending the resources to perform more extensive in vivo trials. The test article described in this study, HM-10/10, is a chimeric high-density lipoprotein created by combining the receptor-binding domain of apolipoprotein E to an apolipoprotein J mimetic. HM-10/10 has shown promise as an anti-inflammatory and subsequently anti-tumorigenic peptide in two of our aforementioned studies assessing tumor development and macular degeneration in mice [11,12]. Before subjecting HM-10/10 to more extensive studies targeting EOCs and CCs, its stability in numerous mediums was assessed in vitro against multiple control articles to investigate its potential efficacy in vivo. HM-10/10 shows good stability in both human plasma and under a range of pH values from 1.3 to 7.4, indicating its promise as an oral pharmaceutical agent. The main hurdle demonstrated by these investigations is its degradation under conditions modeling the small intestine. Further studies should be performed, and are now being planned, to enhance HM-10/10 in order to ameliorate the harsh acidic environment of the stomach while being resistant to the peptidases common to the small intestine. This strategy will hopefully improve the efficacy of HM-10/10 by extending its bioavailability over the desired half-life time course while retaining its inherent low-toxicity profile for use as a safe and effective oral therapeutic drug. This new and enhanced HM-10/10 format holds tremendous potential to add a new therapeutic agent to address the international women’s health crisis of epithelial carcinomas of the ovary and colon.

## 5. Conclusions

The data presented herein highlight the stability characteristics of the native composition of HM-10/10, a novel stable HDL-mimetic peptide with potential anti-cancer activity (e.g., ovary and colon in vitro and in vivo models). The preferred and desirable route of delivery of HM-10/10 is oral (p.o.), and toward that end, HM-10/10 demonstrates promising stability over a broad physiologic pH range which encompasses the environment of the stomach. However, its stability is significantly degraded by conditions simulating the small intestine, likely due to the peptidase-rich conditions therein. Future research will focus on attractive delivery vehicles, including nanoparticle-encapsulated composition of HM-10/10 suitable for withstanding the peptidase-rich environment of the small intestine. The new enhanced format of HM-10/10 is envisioned to be capable of extending its bioavailability while retaining its inherent low-toxicity profile for use as a safe and putatively effective oral therapeutic drug for the potential treatment of EOC and CC.

## Figures and Tables

**Figure 1 ijms-24-10054-f001:**
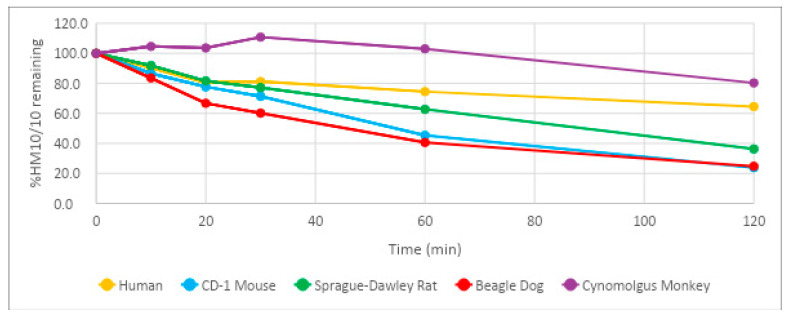
Stability of HM-10/10 in Plasma Matrices. Stability of HM-10/10 in plasma matrices determined by compound concentration measurements over a 120 min time course via triple quadrupole mass spectrometer. All compounds were tested at a concentration of 2 µM.

**Figure 2 ijms-24-10054-f002:**
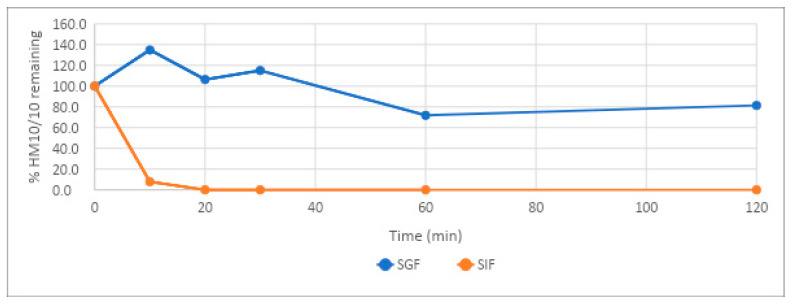
Stability of HM-10/10 in SGF and SIF. Stability of HM-10/10 in simulated gastric fluid (SGF) and simulated intestinal fluid (SIF) determined by compound concentration measurements over a 120 min time course via triple quadrupole mass spectrometer. All compounds were tested at a concentration of 2 µM.

**Figure 3 ijms-24-10054-f003:**
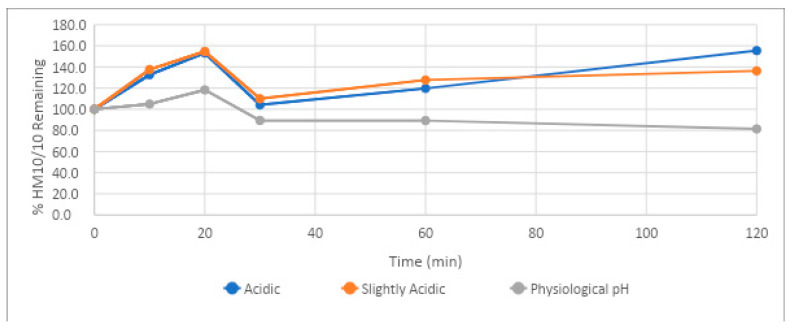
Stability of HM-10/10 under varying pH conditions. Stability of HM-10/10 under varying pH conditions determined by compound concentration measurements over a 120 min time course via triple quadrupole mass spectrometer. Acidic: pH = 1.3, Slightly Acidic: pH = 5.5, Physiological pH: pH = 7.4. All compounds were tested at a concentration of 2 µM.

**Figure 4 ijms-24-10054-f004:**
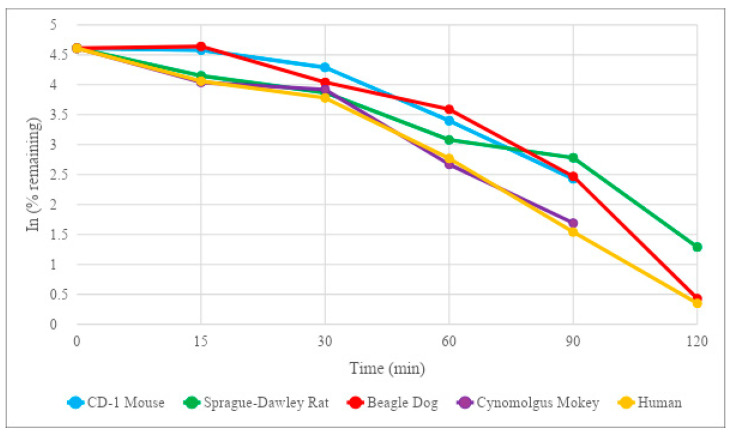
Microsomal Stability Results for HM-10/10 in a Suspension of Liver Microsomes. Microsomal stability results for HM-10/10 in a suspension of liver microsomes (0.5 mg protein/mL) over 120 min. All compounds were tested at a concentration of 2 µM.

**Table 1 ijms-24-10054-t001:** HM-10/10 Stability in Plasma Matrices.

Plasma Matrix Species	Test Compound	T_1/2_	% Remaining at 120 min
Human	HM-10/10	164	64.5
	Lovastatin	646	81.1
	Propantheline	45.6	13.3
Sprague-Dawley Rat	HM-10/10	82.5	36.3
	Lovastatin	8.26	NA ^2^
	Propantheline	4650	95.0
Beagle Dog	HM-10/10	54.6	24.8
	Lovastatin	445	73.2
	Propantheline	246	65.7
Cynomolgus Monkey	HM-10/10	52.7	21.9
	Lovastatin	>120 ^1^	115.2
	Propantheline	663	80.3
CD-1 Mouse	HM-10/10	5.68	23.9
	Lovastatin	14.4	NA ^2^
	Propantheline	70.4	27.0

Test Concentration: 2 μM. ^1^ Half-life could not be determined due to a greater concentration of the compound detected at the 120 min timepoint than at 0 min timepoint. ^2^ NA: Not applicable as compound not detectable at 120 min.

**Table 2 ijms-24-10054-t002:** Stability of HM-10/10 in Simulated Gastric Fluid (SGF) and Simulated Intestinal Fluid (SIF).

Matrix	Test Compound	T_1/2_ (min)	% Remaining at 60 min
Simulated Gastric Fluid (SGF) ^1^	HM-10/10	372	71.8
	Propantheline	522	85.9
Simulated Intestinal Fluid (SIF) ^2^	HM-10/10	2.72	0
	Propantheline	817	95.9

Test Concentration: 2 μM. ^1^ Simulated gastric fluid pH = 1.3. ^2^ Simulated intestinal fluid pH = 5.5.

**Table 3 ijms-24-10054-t003:** Stability of HM-10/10 under different pH conditions.

Matrix	Test Compound	T_1/2_ (min)	% Remaining at 60 min
Acidic (pH 1.3)	HM-10/10	>120 ^1^	155.5
	7-Ethoxycoumarin	808	89.0
Slightly Acidic (pH 5.5)	HM-10/10	>120 ^1^	136.3
	7-Ethoxycoumarin	991	90.0
Physiological pH (pH 7.4)	HM-10/10	433	81.4
	7-Ethoxycoumarin	705	87.5

Test Concentration: 2 μM. ^1^ Half-life could not be determined as the % HM-10/10 remaining after 60 min was greater than it was at 0 min.

**Table 4 ijms-24-10054-t004:** Summary of HM-10/10 Metabolic Stability Results in Cryopreserved CD 1 Mouse, Sprague-Dawley Rat, Beagle Dog, Cynomolgus Monkey, and Human Liver Microsomes (0.5 mg protein/mL).

Species	Compound	T_1/2_ (min)	% Remainingat 120 min	CL_int_ (mL/min/kg)	CL_int_ (L/hr/kg)
CD-1 Mouse	HM-10/10	32.0	0.8	170	10.2
	Verapamil	6.53	0.3	836	50.2
Sprague-Dawley Rat	HM-10/10	27.7	3.7	101	6.09
	Verapamil	8.84	0.08	318	19.1
Beagle Dog	HM-10/10	33.0	1.5	47.2	2.83
	Verapamil	15.7	2.7	99.3	5.96
Cynomolgus Monkey	HM-10/10	21.8	0.8	85.8	5.15
	Verapamil ^a^	<15	0.0	>125	>7.48
Human	HM-10/10	20.3	1.4	61.5	3.69
	Verapamil	13.8	1.2	90.5	5.43

Test Concentration: 2 μM. ^a^: Due to the rapid disappearance of the compound, values are provided for comparative purposes only.

**Table 5 ijms-24-10054-t005:** Summary of HM-10/10 Metabolic Stability Results in Cryopreserved CD 1 Mouse, Sprague-Dawley Rat, Beagle Dog, Cynomolgus Monkey, and Human Hepatocytes (0.5 million cells/mL).

Species	Compound	T_1/2_ (min)	% Remainingat 30 min	CL_int_ (mL/min/kg)	CL_int_ (L/hr/kg)	% Cell Viability (Initial)
CD-1 Mouse	HM-10/10 ^a^	<15	0.7	>1090	>65.5	68.6
	Verapamil	20.2	31.5	810	48.6	68.6
Sprague-Dawley Rat	HM-10/10	5.81	1.0	1260	75.4	90.6
	Verapamil	19.9	23.6	366	22.0	90.6
Beagle Dog	HM-10/10	10.7	16.0	699	42.0	84.4
	Verapamil	45.0	56.0	166	9.94	84.4
Cynomolgus Monkey	HM-10/10	12.1	18.5	465	27.9	91.9
	Verapamil	12.1	17.3	465	27.9	91.9
Human	HM-10/10	5.28	2.5	708	42.5	82.2
	Verapamil	85.8	86.5	43.6	2.62	82.2

Test Concentration: 2 μM. ^a^: Due to the rapid disappearance of the compound, values are provided for comparative purposes only.

**Table 6 ijms-24-10054-t006:** HM-10/10 and Control (Warfarin): Protein Binding in CD-1 Mouse, Sprague-Dawley Rat, Beagle Dog, Cynomolgus Monkey, and Human Plasma (K2EDTA).

Species	Compound	% Protein Free	F_u_ (Unbound Fraction)	% Protein Bound	% Stabilityin Matrix (2.5 h)
		Mean ± SD	Mean ± SD	Mean ± SD	
CD-1 Mouse	HM-10/10	84.7 ± 24.6	0.847 ± 0.246	15.3 ± 24.6	14.6
	Warfarin	4.0 ± 0.2	0.0398 ± 0.0016	96.0 ± 0.2	107.9
Sprague-Dawley Rat	HM-10/10	67.2 ± 9.4	0.672 ± 0.094	32.8 ± 9.4	36.6
	Warfarin	0.7 ± 0.1	0.0073 ± 0.00138	99.3 ± 0.1	104.8
Beagle Dog	HM-10/10	64.8 ± 26.5	0.648 ± 0.265	35.2 ± 26.5	29.2
	Warfarin	3.1 ± 0.4	0.0305 ± 0.0036	96.9 ± 0.4	105.3
Cynomolgus Monkey	HM-10/10	73.6 ± 21.6	0.736 ± 0.216	26.4 ± 21.6	25.9
	Warfarin	0.9 ± 0.1	0.0086 ± 0.00116	99.1 ± 0.1	102.8
Human	HM-10/10	20.3 ± 5.5	0.203 ± 0.055	79.7 ± 5.5	88.5
	Warfarin	0.8 ± 0.1	0.0078 ± 0.00115	99.2 ± 0.1	95.7

Test Concentration: 2 μM.

**Table 7 ijms-24-10054-t007:** Summary of Inhibition of CYP450 Isozymes by HM-10/10 and Control Compounds.

Substrate (CYP Isozyme)	Test Compound	IC_50_ (µM)				% Inhibition at 50 µM				Comments
		No PI ^1^ + NADPH	No PI − NADPH	PI + NADPH	PI − NADPH	No PI + NADPH	No PI − NADPH	PI + NADPH	PI − NADPH	
Phenacetin (1A2)	HM-10/10	ND ^2^	ND	ND	ND	−18.0	−20.7	0.1	−15.8	
	Fluvoxamine	0.201	0.276	0.0410	0.0685	99.2	98.5	100.1	100.5	+TDI ^3^
Buproprion (2B6)	HM-10/10	ND	ND	ND	ND	17.3	25.2	28.4	42.2	
	Ticlopidine	0.369	0.143	0.217	0.327	98.2	98.8	99.6	97.9	No TDI
Amodiaquine (2C8)	HM-10/10	ND	ND	ND	ND	18.7	14.0	9.0	11.4	
	Quercetin	2.10	1.90	2.81	1.96	100.5	100.9	100.8	100.5	No TDI
Diclofenac (2C9)	HM-10/10	ND	ND	ND	ND	8.6	0.4	1.7	0.5	
	Sulfaphenazole	0.515	0.389	0.610	0.464	99.2	99.1	99.7	99.9	No TDI
Mephenytoin (2C19)	HM-10/10	ND	ND	ND	ND	−26.4	−9.4	−6.0	19.9	
	Omeprazole	4.77	5.39	2.54	5.00	92.9	93.1	95.8	90.7	+TDI
Dextromethorphan (2D6)	HM-10/10	ND	ND	ND	ND	6.0	17.2	17.0	19.7	
	Paroxetine	0.518	0.573	0.0873	0.593	99.0	99.1	100.1	99.0	+TDI
Midazolam (3A4)	HM-10/10	ND	ND	ND	ND	6.6	19.2	13.9	29.5	
	Mifepristone	2.88	2.70	0.915	2.91	86.4	85.4	96.8	86.7	+TDI
Testosterone (3A4)	HM-10/10	ND	ND	ND	ND	6.9	−0.7	11.3	32.2	
	Mifepristone	2.85	4.69	0.298	4.76	93.3	93.5	100.2	91.4	+TDI

^1^ PI = preincubation with microsomes (enzyme source), ^2^ ND = not determined, ^3^ TDI = time-dependent inhibition.

**Table 8 ijms-24-10054-t008:** HM1010: Cytochrome P450 (CYP1A2, CYP2B6, and CYP3A4) Induction (Enzyme Activity: Fold-Increase) in Three Lots of Single-Donor Human Hepatocytes.

CYP450		Concentration	Fold Increase (Enzyme Activity)				Average
Isozyme	Compound	(µM)	Lot HH1086	Lot HH1142	Lot HH1144	Mean	% Stability
1A2	HM1010	2	0.682	1.33	1.73	1.25	0.6
		10	1.88	5.19	3.32	3.46	0.7
		25	2.46	5.98	3.65	4.03	0.3
	Omeprazole	50	15.4	10.3	11.3	12.3	NA
2B6	HM1010	2	3.18	2.82	3.00	3.00	0.6
		10	4.01	6.36	4.52	4.96	0.7
		25	2.80	9.29	6.73	6.27	0.3
	Phenobarbital	1000	11.8	21.4	7.23	13.5	NA
3A4	HM1010	2	2.06	1.59	1.35	1.67	0.6
		10	2.20	2.12	1.83	2.05	0.7
		25	2.07	2.19	2.15	2.14	0.3
	Rifampicin	20	9.15	11.7	33.6	18.2	NA
Note:	Fold increase ≥ 2 is considered induction						
NA:	Not applicable						

Fold Increase = Induced Mean Area Ratio/No Induction Mean Area Ratio.

**Table 9 ijms-24-10054-t009:** mRNA Fold Induction of HM-10/10 with Three Different CYP450 Isoenzymes.

Compound ID			Fold Increase (mRNA Expression) ^1^ of Hepatocyte Lots				
Inducer	Concentration of Assay	CYP450	Lot HH1086	Lot HH1142	Lot HH1144	Mean of 3 Lots	Comments
HM-10/10	2	1A2	1.69	2.24	1.81	1.91	No Induction Observed
HM-10/10	10	1A2	2.75	3.24	1.75	2.58	Induction Observed
HM-10/10	25	1A2	2.74	3.01	1.13	2.29	Induction Observed
HM-10/10	2	2B6	2.03	9.76	2.86	4.88	Induction Observed
HM-10/10	10	2B6	2.54	7.08	3.33	4.32	Induction Observed
HM-10/10	25	2B6	1.26	5.04	1.92	2.74	Induction Observed
HM-10/10	2	3A4	7.87	7.73	1.08	5.56	Induction Observed
HM-10/10	10	3A4	9.17	7.79	1.03	6.00	Induction Observed
HM-10/10	25	3A4	6.99	10.0	1.02	6.00	Induction Observed
Controls			Fold Increase (mRNA expression) ^1^ of Hepatocyte Lots				
Omeprazole	50	1A2	24.5	13.6	19.0	19.0	Induced as Expected
Phenobarbital	1000	2B6	15.5	22.3	7.54	15.1	Induced as Expected
Rifampicin	20	3A4	69.7	139.0	133.0	114	Induced as Expected

^1^ Fold increases in enzyme activity ≥2.0-fold that of the vehicle control (PBS or DMSO) were considered induction.

**Table 10 ijms-24-10054-t010:** Optimized Parent Ion/Product Ion Transitions.

Compound	Q1/Q3
HM-10/10_1	508.5/85.0
Feng#1 SIL (IS)	510.2/86.2
HM-10/10_2	508.3/565.8 (for SGF and SIF)
Carbutamide (IS)	272.1/156.0
Lovastatin	405.1/199.0
Glyburide (IS)	494.0/168.9
Propantheline	367.7/181.2
Carbutamide (IS)	272.1/156.0
HM-10/10_3	508.3/602.6 (for buffer)
Carbutamide (IS)	272.1/156.0

IS: Internal standard. SFG: Simulated gastric fluid. SIF: Simulated intestinal fluid.

## Data Availability

Data supporting reported results can be found at Susan Dearborn; Charles River Laboratories International, Stone Ridge, NY 12484, USA; susan.dearborn@crl.com.

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
