# Peer review of "Stability Characterization of the Novel Anti-Cancer HM-10/10 HDL-Mimetic Peptide"

_ijms, 2023, doi:10.3390/ijms241210054_

Round 1

Reviewer 1 Report

The authors described "Pharmacokinetic/Pharmacodynamic Characterization of the Novel Anti-Cancer HM-10/10 HDL-Mimetic Peptide" in this manuscript.

I undestand that the details of PK data with HM-10/10 HDL-Mimetic Peptide should be informative for many readers. However, the story of this manuscript is just to identify the issues of peptide stability in plasma, not to provide the actual data of the solution. I felt that this manuscript should be reconsidered its style. It is not worth to reviewing its novelity of the author's approach to solve the peptide PK issues. Just assessment of peptide with in vitro and in vivo, is still the introduction of typical manuscript.

Quality of English Language in this manuscript is fine.

Reviewer 2 Report

1. Tables 2 and 3 can be combined or generally given in the text, they contain little information and weigh down the text of the manuscript.

2. Table 4, 5, etc. is a column with the same concentration necessary? The concentration can be indicated in the note to the table or in the text.

3. In Fig.1, 2, etc., give the intervals of variation. In how many parallels was the study conducted?

4. Line 395 - maybe control compounds?

Reviewer 3 Report

The manuscript deals with the stability assessment of HM-10/10as an anti-cancer and HDL mimetic peptide. The topic is interesting but the manuscript requires substantial revisions.

The origin, sources and purity of HM-10/10 peptide is not reported.

It is not clear how the stability of the peptide was assessed in terms of chemical quantification. All analytical information about the results and method used should be added as supplementary material.

Paragraph 3.1 and 3.2 and others: According to which reference the stability of HM-10/10 is considered “good”?

Format and appearance of tables and graphs should be improved.

Conclusions should be revise and they must better reflect the findings of the present study.

Line 672 The study does not demonstrate the therapeutic effectiveness toward epithelial adenocarcinomas and not toxicity.

The title does not reflect the content of the manuscript. It has been not performed a pharmacokinetic/pharmacodynamic characterization of the HM-10/10 but it is focused on the stability of the peptide at different biological conditions.

Introduction must be improved and the aim of the manuscript (line 70-85) should be modified since a pharmacokinetic/pharmacodynamic characterization was not performed, only a stability study on HM-10/10 peptide.

The use of peptide in anticancer therapy and the potential use as well as the HDL-mimetic activity of HM-10/10 peptide should be better highlighted in the introduction.

The origin of plasma and liver microsomes as well as other biologics is not detailed and reported.

Round 2

Reviewer 1 Report

As the second review, the authors corrected criticized points. I agree that this manuscript now can be publicated in this journal.

Author Response

Thank you.

Reviewer 3 Report

The manuscript has been improved after the revisions from the authors.

However, some points indicated by the reviewer has not been addressed.

1) Previous comment from Reviewer: “The origin, sources and purity of HM-10/10 peptide is not reported”.

The authors have specified that “Synthesis of HM-10/10 adhered to FDA GLP compliance protocols for purity and quality assurance” in the revised manuscript. However, it is not reported how the peptide was synthesized or from whom the peptide was provided. Without this information, the material used have not specifications and it sounds not scientific.

2) Previous comment from Reviewer: “It is not clear how the stability of the peptide was assessed in terms of chemical quantification. All analytical information about the results and method used should be added as supplementary material”.

This point has not been addressed by the authors. I understand that the analytical method is reported in the paragraph 2.11 Bioanalysis but how the quantification of the peptide and degradation products were performed at the different conditions from an analytical point of view should be better reported and addressed.

3) Previous comment from Reviewer: The use of peptide in anticancer therapy and the potential use as well as the HDL-mimetic activity of HM-10/10 peptide should be better highlighted in the introduction.

The introduction must be implemented with this information

4) The origin of plasma and liver microsomes as well as other biologics is not detailed and reported.

This information is still missing

I suggest answering point-by-point to reviewer comments in addition to preparing a cover letter to editors.

Round 3

Reviewer 3 Report

The manuscript is suitable for publication